# Yeast Lipid Produced through Glycerol Conversions and Its Use for Enzymatic Synthesis of Amino Acid-Based Biosurfactants

**DOI:** 10.3390/ijms24010714

**Published:** 2022-12-31

**Authors:** Dimitris Karayannis, Seraphim Papanikolaou, Christos Vatistas, Cédric Paris, Isabelle Chevalot

**Affiliations:** 1Reaction and Chemical Engineering Laboratory, University of Lorraine, F-54000 Nancy, France; 2Department of Food Science & Human Nutrition, Agricultural University of Athens, 75 Iera Odos, 11855 Athens, Greece

**Keywords:** biosurfactant, microbial lipid, aminoacylases, N-acylation, specificity factor

## Abstract

The aim of the present work was to obtain microbial lipids (single-cell oils and SCOs) from oleaginous yeast cultivated on biodiesel-derived glycerol and subsequently proceed to the enzymatic synthesis of high-value biosurfactant-type molecules in an aqueous medium, with SCOs implicated as acyl donors (ADs). Indeed, the initial screening of five non-conventional oleaginous yeasts revealed that the most important lipid producer was the microorganism *Cryptococcus curvatus* ATCC 20509. SCO production was optimised according to the nature of the nitrogen source and the initial concentration of glycerol (Glyc0) employed in the medium. Lipids up to 50% *w*/*w* in dry cell weight (DCW) (SCO_max_ = 6.1 g/L) occurred at Glyc0 ≈ 70 g/L (C/N ≈ 80 moles/moles). Thereafter, lipids were recovered and were subsequently used as ADs in the N-acylation reaction catalysed by aminoacylases produced from *Streptomyces ambofaciens* ATCC 23877 under aqueous conditions, while *Candida antarctica* lipase B (CALB) was used as a reference enzyme. Aminoacylases revealed excellent activity towards the synthesis of acyl-lysine only when free fatty acids (FAs) were used as the AD, and the rare regioselectivity in the α-amino group, which has a great impact on the preservation of the functional side chains of any amino acids or peptides. Aminoacylases presented higher α-oleoyl-lysine productivity and final titer (8.3 g/L) with hydrolysed SCO than with hydrolysed vegetable oil. The substrate specificity of both enzymes towards the three main FAs found in SCO was studied, and a new parameter was defined, viz., Specificity factor (Sf), which expresses the relative substrate specificity of an enzyme towards a FA present in a FA mixture. The Sf value of aminoacylases was the highest with palmitic acid in all cases tested, ranging from 2.0 to 3.0, while that of CALB was with linoleic acid (0.9–1.5). To the best of our knowledge, this is the first time that a microbial oil has been successfully used as AD for biosurfactant synthesis. This bio-refinery approach illustrates the concept of a state-of-the-art combination of enzyme and microbial technology to produce high-value biosurfactants through environmentally friendly and economically sound processes.

## 1. Introduction

Surfactants are compounds that, due to their amphiphilic nature, alter the interfacial composition and rheology, reducing the interfacial tensions and increasing the interfacial area [1]. These surface-active molecules have wide applications in the food, pharmaceutical and cosmetic industries. Biosurfactants, a class of green and sustainable surfactants are naturally synthesised from microorganisms (bacteria, fungi, yeast and microalgae) or they can be generated from the association of a polar amino acid (hydrophilic moiety) and a non-polar long-chain compound (hydrophobic moiety) from renewable sources [2]. The global biosurfactants market is projected to grow from $3.96 billion in 2021 to $5.71 billion in 2028 at a CAGR of 5.4% in the forecast period, 2021–2028 [3]. To date, amino acid-based surfactants are industrially produced via a chemical route, called the Schotten-Baumann reaction, which has been associated with major environmental drawbacks (e.g., salted wastes, acyl chlorides, organic solvents, etc.) [4].

These amino acid-based surfactants have a low degree of toxicity, low haemolytic activity and are easily biodegradable [2,5]. They are widely applied in the pharmaceutical, food and cosmetic industries [6] due to their excellent emulsifying and antimicrobial activities [7]. Furthermore, the great variety of amino acids and FAs translates into a potential for diverse biosurfactant development, exhibiting various properties [8]. Furthermore, it is well established that some peptides present various biological activities [9] (e.g., antioxidant, opioids, anti-hypertensive, anti-tumour), although at the same time they lack stability towards endogenous proteolysis and in most cases their polarity can limit their passage through biological membranes [10]. As a result, acylated derivatives of peptides can increase the bioavailability and permeability of the original peptides [11]. For instance, Asada et al. (1994) have demonstrated that mono-acylation can improve insulin stability inside the human body [12]. Amino acid acylation can occur in amino (N-α/ε-acylation) [13] and/or carboxylic group (O-acylation) [14,15].

Microbial lipids (single-cell oils and SCOs), viz., lipids that are produced from single-celled entities (the oleaginous microorganisms, which are microbial cells that when culture conditions are adequate, can accumulate lipids to more than 20%, *w/w*, of their dry cell mass during growth on glucose or similarly metabolised compounds), are hydrophobic compounds that present a great academic and industrial interest [16,17,18]. Although the production price of these lipids is almost always higher than one of the traditional commodity oils and fats [16,18], these lipids can be used as replacements for several types of expensive fatty materials that may be scarcely found in the Plant or Animal Kingdom (i.e., cocoa-butter, shea-butter, borage oil, infant milk lipid, lipids from deep-sea fish, etc.) [17,18,19,20]. On the other hand, the latter developments and economic events (i.e., the economic crisis of the last years accompanied by the war in Eastern Europe, etc.) resulted in a steep increase in the cost of several types of agro-industrial/agricultural products and edible commodities such as the plant-origin lipids (sunflower oil, soybean oil, etc.). Due to their resemblance in the fatty acid composition with the various types of plant-based commodity oils (soybean oil, rapeseed oil, olive pomace oil, etc.), SCOs do not have competition with food products, and, thus, these lipids could be used as unconventional replacements for the mentioned plant-based fatty materials in their utilization as substrates implicated in the production of biodiesel or several oleochemical and (bio)-chemical industrial processes [18,21].

Enzymatic reactions seem to be one of the most attractive alternatives to chemical processes, as they are generally performed under mild experimental conditions (e.g., temperature, pressure), generate less waste [22] and provide elevated reaction rates, high specificity and regioselectivity to acylation reactions [23]. In biocatalytic processes, there is usually neither a need for functional-group activation, protection and deprotection steps nor a need for high-pressure reactors and other types of costly equipment [24], which makes the bioprocess more step-economical [25].

Concerning enzymatic N-acylation reaction, Soo et al. (2004) reported the ability of *Rhizomucor miehei* to perform lysine acylation of the *ε*-amino group in the organic solvent, where the conversion yield of *ε*-palmitoyl-lysine and *ε*-oleoyl-lysine reached 16% and 33%, respectively [4]. Lipases are widely known as a useful family of enzymes since they can transfer the acyl group in organic solvents [26], in solvent-free reactions and in ionic liquids [15,27]. It has been reported by Husson et al. (2008), that carnosine, a polar dipeptide, was successfully N-acylated by oleic acid in 2-Methyl-2-butanol (M2B2), and a yield of 39% was obtained due to substrate dispersion improvement by the application of a high pressure on the reaction medium [28].

More recently, enzymes such as aminoacylases have been described for their ability to catalyse N-acylation reactions in aqueous mediums, unlike lipases whose thermodynamic equilibrium dictates the product’s hydrolysis instead of its synthesis. Koreishi et al. (2006) reported a novel acylase from *Streptomyces mobaraensis*, which was able to catalyse the synthesis of N-fatty-acyl-amino acids and N-fatty-acyl-peptides in an aqueous/organic biphasic system [29]. When lysine was used, there was an enzyme specificity towards the ε-amino group of lysine, as the ε-lauroyl-lysine yield was 10-fold higher than α-lauroyl-lysine. In another major study, it is reported that a recombinant ε-lysine acylase from *S. mobaraensis* was able to synthesise lauroyl-lysine in an aqueous medium, and its yield was close to 100% after 6 and 9 h of reaction for 50 and 100 mM lauric acid, respectively [30].

Novel aminoacylases deriving from *Streptomyces ambofaciens* have recently been identified by other authors [31,32]. They revealed excellent activity toward the synthesis of acyl-amino acids under aqueous conditions and regioselectivity in the *α*-amino group. Dettori et al. (2017) presented this rare ability of crude extract from *S. ambofaciens* to catalyse the N-*α*-acylation, which has a great impact on the preservation of the functional side chains of any amino acids or peptides, whereas CALB catalysed the N-*ε*-acylation reaction [31]. Bourkaib et al. (2020) reported the acylation with the undecylenic acid of the 20 proteogenic amino acids. The highest yields of the N-α-acylation reactions were obtained with positively charged amino acids (lysine and arginine), while no reaction was obtained with the negatively charged ones (aspartic and glutamic acids) [32].

This aminoacylases regioselectivity can have several benefits of product properties, as the ε-amino group preserves its functional properties. Pérez et al. (2009), who studied the influence of the N-acylated group of the lysine methyl and ethyl-ester on the antimicrobial and haemolytic activity [33], demonstrated that the N-acylation on *ε*-position decreased considerably the antimicrobial activity and increased the haemolytic one, as compared to what was observed when acylation occurred at *α*-position. This difference in antimicrobial activity is attributed to the pKa associated with the protonated amino group of the polar heads. These results have increased the interest in the N-α-acylation of amino acids, especially when using peptides or amino acids with more than one free amino group for acylation, such as lysine.

As previously mentioned, microbial lipids with a FA composition similar to that of various commodity oils can also have many benefits to the end products (biosurfactants) as well as the environment. These lipids are not food competitors, but they can be produced during the growth of oleaginous microorganisms on wastes and byproducts [34,35,36]. Specifically, with the very high expansion of biodiesel production in the last years, glycerol (which is, as mentioned, the main side-product of the biodiesel production process) is accumulated in tremendous quantities in the market volume, evidently resulting in a very important drop in the price of this compound [37,38]. Therefore, it is not surprising that the topic of the valorisation of this residue is amongst the “hottest” ones for the biotechnological industries, with a plethora of metabolites having been produced in the laboratory- and pilot-scale experiments using this residue as starting material in numerous bioprocesses [18,34,35,36,37,38]. The aim of the present work was the enzymatic production of amino acid-based surfactants from a microbial oil in aqueous media. To achieve this, biodiesel-derived glycerol, the principal side-product of biodiesel production, was employed as substrate and a selection amongst 5 non-conventional yeast strains was first carried out under culture conditions favouring the production of microbial lipids (viz., in trials performed under glycerol-excess and nitrogen-limiting conditions). The most promising amongst these yeast strains was further studied and the production of microbial lipids by this strain was optimised as regards the nature of the nitrogen source used and the initial concentration of glycerol employed in the medium. Thereafter, lipids of this strain were recovered and were subsequently used as ADs in the N-acylation reaction catalysed by aminoacylases produced from *S. ambofaciens* ATCC 23877 culture. At the same time, the productivity, substrate specificity and regioselectivity of these aminoacylases towards the main FAs found in microbial and rapeseed oil were compared. CALB, which can only catalyse N-acylation in a non-aqueous medium, and rapeseed oil, an edible vegetable oil with a FA composition similar to *C. curvatus* oil, were used as reference enzymes and reference oils, respectively. Therefore, the goal of the present study was to propose a bio-refinery approach, combining microbial and enzymatic processes in which a low-cost carbon source (crude glycerol) was employed as a microbial substrate by oleaginous yeasts, and the subsequent fatty material produced was enzymatically converted into high-value biosurfactant-type molecules through green conversions.

## 2. Results and Discussion

### 2.1. Production and Characterization of Microbial Lipids

#### 2.1.1. Lipid Production by Yeast Cultures Growing on Biodiesel-Derived Glycerol

In the first part of this study, 5 wild-type yeast strains were cultivated on biodiesel-derived glycerol, (at Glyc0 ≈ 50 g/L, nitrogen sources yeast extract at 1.0 g/L and peptone at 2.0 g/L) and the obtained results are seen in Table 1. All microorganisms tested produced noticeable DCW quantities and satisfactorily assimilated glycerol from the medium. Despite the nitrogen-limited conditions imposed, that favoured the biosynthesis and production of SCOs [18,19,20] amongst the 5 strains tested, only 2 of them (namely *R. toruloides* NRRL Y-6984 and *C. curvatus* ATCC 20509) showed lipid in DCW values that were ≥20% *w/w*, a value-threshold characterising the oleaginicity of the implicated microbial strains [17]. Interestingly, the microorganism *C. curvatus* NRRL Y-1511 did not produce sufficient lipid quantities (lipid in DCW values ≤ 6.0% *w/w*), although in earlier investigations (in which sugars and not glycerol had been employed as substrates) it proved capable of producing some SCO quantities inside its cells [39]. Moreover, all microorganisms produced variable quantities of endopolysaccharides. For the tested strains, *M. pulcherima* FMCC Y2 seemed to increase the content of polysaccharides per DCW, which finally reached a quite high intra-cellular content (viz., 43% *w/w*, in accordance with the results reported for other *Metschnikowia* sp. strains [40]). Moreover, the strain *Rhodotorula* sp. FMCC Y76 seemed to maintain quite high intra-cellular polysaccharides quantities (36–38% *w/w*), whereas lower cellular lipid quantities were recorded simultaneously. Finally, all other remaining strains (i.e., *R. toruloides* NRRL Y-6984, *C. curvatus* NRRL Y-1511 and *C. curvatus* ATCC 20509) presented non-negligible intra-cellular polysaccharides quantities per unit of DCW at the relatively earlier growth steps, that decreased as the cultures proceeded. This result, although not in full agreement with the literature (endopolysaccharides as intra-cellular lipids request carbon-excess and nitrogen-limited conditions in order to be synthesised, and therefore enhanced quantities of polysaccharides per unit of DCW would be anticipated to occur at the late fermentation steps and after nitrogen deprivation; [18,41]), has already been reported in previous studies; in fact, it has been demonstrated that *C. curvatus* and *R. toruloides* cultures in culture conditions enabling the de novo lipid production process have presented elevated endopolysaccharides in DCW values at the earlier growth phases, with the presence (or barely with the absence) of nitrogen from the medium, with these values being depleted as growth proceeded [35,39,40], in accordance with the results reported in the present study. From all results presented, it may be assumed that the highest lipid production in both absolute (in g/L) and relative (in g of lipid per g of DCW) values was recorded by the microorganism *C. curvatus* ATCC 20509.

In the next step and given that the best results concerning SCO production were presented by *C. curvatus* ATCC 20509, this microorganism was further studied as regards its lipid accumulation capacities when several types of (organic and inorganic) nitrogen sources were added into the growth medium. Previous investigations, dealing with studies of the strain *R. toruloides* CBS 14 on glucose-based media in shake-flask nitrogen-limited trials, have demonstrated that the nature of the nitrogen source (taking into consideration that total nitrogen quantity added into the medium was always constant) played a crucial role in the process of lipid production [41]. In the present study, various inorganic and organic nitrogen combinations (in all cases and irrespective of the principal nitrogen source used, yeast extract had always been added into the medium at 1.0 g/L; see “Section 3.2”) were employed, and, in accordance with the literature [17,41], the employed nitrogen source played a crucial role upon SCO production in *C. curvatus* ATCC 20509. Interestingly, the utilisation of inorganic nitrogen sources (i.e., ammonium sulphate and potassium nitrate) did not have a systematic effect on the lipid accumulation process; for instance, the employment of ammonium sulphate shifted the metabolism towards the production of total biomass and lipid-free biomass and to a lesser extent towards total lipids (maximum lipids in DCW ≈ 18% *w/w*), while the utilisation of potassium nitrate resulted in lower total DCW quantities obtained, that contained higher SCO quantities (lipids in DCW up to 39% *w/w*). The utilisation of organic nitrogen sources was not a prerequisite for significant production of lipids in terms of lipids in DCW values, in contrast to the results reported for *R. toruloides* CBS 14, where, by far, the most suitable nitrogen sources were the amino acids glutamate and arginine and the organic compound of urea [41]. In contrast, in our study, neither the utilisation of urea nor the utilisation of yeast extract seemed to enhance the process of lipid accumulation. By taking into consideration the production of total DCW (in g/L), the production of lipids (in g/L), and the value of lipids accumulated per unit of total DCW, it may be indicated that the best nitrogen combination was that of yeast extract (added at 1.0 g/L) and peptone (added at 2.0 g/L) (see Table 2).

Given that the most suitable combination of nitrogen sources amenable to boost the process of lipid accumulation in *C. curvatus* ATCC 20509 was that of yeast extract (1.0 g/L) and peptone (2.0 g/L), in the next and final step of lipid production optimisation it was decided to study the effect of the Glyc0 concentration (and also the effect of the initial C/N molar ratio) when constant initial nitrogen concentration (yeast extract at 1.0 g/L and peptone at 2.0 g/L) was added into the medium. Recent investigations employing the currently used *C. curvatus* ATCC 20509, during its cultivation on second cheese-whey supplemented with cheese-whey lactose, have demonstrated the very high significance of the initial carbon source (lactose) concentration and the initial C/N molar ratio upon the bioprocess [42]. Therefore, in the current investigation, different Glyc0 concentrations (≈30, ≈50 and ≈70 g/L) were added into the medium at constant nitrogen concentration, and the effect of both the increasing Glyc0 concentrations and initial C/N molar ratios was studied (Table 3). From the obtained results, it is observed that the more the Glyc0 concentration increased, the less the maximum DCW values were recorded. This could be due to substrate (glycerol) inhibition exerted towards the strain, due to the increased Glyc0 concentrations found in the medium, in accordance with the results reported by Meesters et al. (1996) [43]. Moreover, from the obtained results, it can be assumed that the more growth was performed under carbon-excess conditions (high initial C/N molar ratios) the more SCOs were produced inside the yeast cells (lipids increased at both absolute; g/L, and relative; g of lipid per g of DCW, values). Specifically, at Glyc0 ≈ 70 g/L (C/N ≈ 80 moles/moles), lipids up to 50% *w/w* in DCW were produced (SCOmax = 6.1 g/L), although total DCW values were smaller compared to the trials with the lower Glyc0 concentrations imposed (Table 3). Finally, as in most of the previously presented trials, the relative values of polysaccharides (g/g of total DCW) presented increased values at the earlier growth steps, decreasing significantly as the growth proceeded, simultaneously with the increase of lipids in DCW values, in accordance with literature reports for a few oleaginous yeast species [35,39,40]. One of the characteristic cases (growth at Glyc0 ≈ 70 g/L; C/N ≈ 80 moles/moles) is illustrated in Figure 1a,b.

#### 2.1.2. Yeast Lipid Analysis

All screened strains were analysed with regards to the FA composition of their total lipids at the stationary growth phase (Table 4). In agreement with most of the literature reports [17,18,19], the principal FAs found in yeast lipids were mainly oleic acid (C18:1), palmitic acid (C16:0) and linoleic acid (C18:2), whereas there are lower quantities of stearic acid (C18:0), palmitoleic acid (C16:1) and α-linolenic acid (C18:3) (Table 5).

Long chain and rarely found in the plants or fish poly-unsaturated FAs (e.g., EPA, DHA, etc.) were not detected in high concentrations, since these compounds are the principal storage lipophilic compounds in oleaginous fungi and algae [19,20,44,45]. These rarely found FAs can be produced in significant quantities inside the yeast cells only after genetic modifications have been performed [20,45]. Equally, low quantities of stearic acid have been identified and quantified inside the yeast cells, with this event being the limiting step for the microbial production of substitutes for cocoa butter or other saturated exotic fats [17,18]. Total FA composition analysis of the cellular lipids of *C. curvatus* ATCC 20509 presented some small differentiations related to the utilisation of different nitrogen sources and/or the Glyc0 concentration imposed into the medium (Table 5 and Table 6), with these differentiations not following any systematic trend.

### 2.2. N-Acylation Activity and Regioselectivity of CALB and Aminoacylases

CALB and aminoacylases were studied for their ability to catalyse the N-acylation reaction of lysine with oil-derived FAs, such as a microbial and a vegetable oil. Their regioselectivity towards α or ε position of lysine and their productivity were investigated. Free oleic acid was used as a reference AD with both enzymes, as the main product studied was oleoyl-lysine. Lipids of *Cryptococcus curvatus* ATCC 20509 culture at Glyc0 ≈ 70 g/L (C/N ≈ 80 moles/moles) were used as microbial oil, while rapeseed oil was used as a reference vegetable oil.

#### 2.2.1. N-Acylation of Lysine Using Free and Oil-Derived AD by CALB in Non-Aqueous Medium

Immobilised lipase B derived from *Candida antarctica* yeast, named *Candida antarctica* lipase B (CALB) has been selected as a reference enzyme for its ability to perform the N-acylation of lysine in a non-aqueous medium. The presence of water in the reaction medium can affect the thermodynamic equilibrium of the enzyme, as it drives its activity to product hydrolysis rather than synthesis. Therefore, the absence of water in the reaction medium is of foremost importance for the development of conditions in which the synthetic (N-acylation) activity of CALB will be favoured. In this experimental approach, a dehydrated organic solvent was used to enhance the synthetic activity. As has been already reported [46], the enzyme’s regioselectivity navigates acylation to the ε-amino group of lysine.

The results from HPLC analysis indicated that CALB was able to synthesise acyl-lysine, as N-acylation was performed regardless of the type (free FAs or in triacylglycerol structure) of AD. Henceforth, oleoyl-lysine will be considered as the reference product since it was found in all different mixtures and was the major product (Figure 2a).

Furthermore, in order to discriminate products that occurred from different N-acylation positions of lysine, ε- and α-oleoyl-lysine were analysed as standards. The qualification was based on their Mass Spectrometry (MS2) profiles, which were generated by the fragmentation of the parent ions. Fragmentation of ε-oleoyl-lysine led to one major daughter ion (*m/z* = 348) corresponding to [(Oleoyl-lysine) − H_2_O − COOH + H^+^]. The enzymatic reaction of CALB was selective, favouring the production of ε-oleoyl-lysine, verifying the results of a previous study [46] (Figure 3). Figure 2a demonstrates the kinetic of product synthesis and the different ε-oleoyl-lysine concentrations and productivity, depending on the oleic acid’s origin. In the cases of *Cryptococcus curvatus* ATCC 20509 oil (Cu) and rapeseed oil (Rap) product concentration was around 14 g/L, while productivity was 0.57 g/L/h and 0.43 g/L/h respectively after 24h. On the contrary, in the case of free oleic acid, product concentration was 12.5 g/L, but a significantly lower productivity was achieved, namely, 0.29 g/L/h after 24 h. It is of major interest to focus on the level of productivity of this bioprocess, as Cu has proved to be the most suitable substrate for enzymatic acylation by CALB. The plateau was observed at approximately 14 g/L, corresponding to an 80% yield of lysine to N-ε-oleoyl-lysine. The lower product titre and productivity in the case of oleic acid may be due to the presence of water in the medium. As each mole of product generates one mole of water, the thermodynamic equilibrium of CALB can be driven from synthesis (e.g., organic medium) to hydrolysis (aqueous medium). This phenomenon is not observed when FAs are found in triacylglycerol structure (e.g., Cu, Rap) since, each time a FA is liberated, one mole of water is consumed.

#### 2.2.2. N-Acylation of Lysine Using Free and Oil-Derived AD by Aminoacylases, in Aqueous Medium

After the production and the partial purification of aminoacylases, their final protein concentration was measured at 9 g/L by Bradford assay. Recently isolated aminoacylases from *Streptomyces ambofaciens* ATCC 23877 were selected for their ability to perform N-acylation of α amino group of amino acids or peptides [31]. Fragmentation of α-oleoyl-lysine led to one major daughter ion (*m/z* = 375.4) corresponding to [(Oleoyl-lysine) − 2H_2_O + H^+^] and one minor daughter ion (*m/z* = 348.35) corresponding to [(Oleoyl-lysine) − H_2_O − COOH + H^+^]. The experimental results, presented in Figure 4, confirm that the enzymatic reaction by aminoacylases was selective, favouring the production of α-oleoyl-lysine. At first, the effect of the initial form of AD on the ability of aminoacylases to carry out the N-acylation of lysine was investigated. More specifically, in one case, the ADs were in free form (hydrolysed *C. curvatus* oil), and in the other, they were in a triacylglycerol structure (*C. curvatus* oil). The results of HPLC analysis indicated that only when AD was in free form N-acylation occurred and that a non-hydrolysed oil cannot be used by aminoacylases as an acyl carrier in order to perform an N-acylation. This finding determined the approach adopted to develop the next bioprocesses with this enzyme. Thus, a stepwise process was required, in which microbial oil from *C. curvatus* (Cu) and rapeseed oil (Rap) were hydrolysed to release the free FAs from the triacylglycerol structure and make them available for the acylation reaction by aminoacylases. The hydrolysis step was also performed enzymatically, with CALB in an aqueous medium. In the second step, hydrolysed oils were used as AD for the N-α-acylation of lysine by aminoacylases. The synthesis of α-oleoyl-lysine was successfully performed, when the ADs were found in hydrolysed *C. curvatus* oil (H-Cu), in hydrolysed rapeseed oil (H-Rap) and in free oleic acid (Figure 2b). The highest concentration of α-oleoyl-lysine was around 8.3 g/L and occurred in the cases of H-Cu and free oleic acid at 120 h, while productivity at 24 h was 0.16 g/L/h and 0.08 g/L/h, respectively. Product concentration with H-Rap was only 2.2 g/L. It is very important to highlight the fact that a hydrolysed microbial oil (H-Cu) gave almost the same product concentration and three times higher productivity compared to free oleic acid and almost four times higher product concentration than a vegetable oil (H-Rap). This is a very promising result, considering that the bioprocess has not been yet optimised, even though its productivity is much lower than that of CALB. Therefore, an approach for increased activity of aminoacylases, as a metalloenzyme, could be the addition of cobalt ions to the reaction medium [32].

### 2.3. Substrate Specificity of CALB and Aminoacylases towards FAs from Oils or Mixtures

In both microbial and vegetable oil used, there was more than one FA available for CALB or aminoacylases to perform the N-acylation of lysine, but the products of such reactants could not be properly detected and quantified by the HPLC analysis, due to the lack of standards (e.g., a-linoleoyl-lysine) in the market and their very similar retention times. Thanks to the determination of the FA composition of both Cu and Rap (Table 7), the possible N-acylation products of lysine can be deduced. A qualitative and semi-quantitative analysis, such as HPLC-MS was therefore applied for the accurate identification of major products namely N-α/ε-palmitoyl-lysine, N-α/ε-oleoyl-lysine and N-α/ε-palmitoyl-lysine. The aim of this investigation was to understand the substrate specificity of both enzymes towards FAs derived from a microbial oil (Cu), a vegetable oil (Rap) and tailor-made mixtures of free FAs in different molar ratios, as well as the possible effect of the hydrolysis step that took place in the case of aminoacylases.

In enzymatic assays with CALB, 3 different oils were used as AD carriers; Cu, Rap and Iso-mix, their FA compositions are shown in Table 7. Iso-mix is an equimolar mixture of the 3 main FAs, found in Cu and Rap, that is, 1/3 of palmitic acid, 1/3 of oleic acid and 1/3 of linoleic-acid. In Table 8 each product’s Mass Area and its % in the sum of products are presented. In the case of Iso-mix, where there is an equal quantity of each AD, ε-linoleoyl-lysine was the major product with 45.3%, in contrast to Cu and Rap, where ε-oleoyl-lysine was the main product, having 66.3% and 51.2% respectively. At this point, it is of high importance to consider the FA composition of both oils used (% of each AD), in order to understand the substrate specificity of the enzyme, which was expected to verify Iso-mix results, where all ADs were in equal concentration and linoleic acid gave the highest product (ε-linoleoyl-lysine). The specificity factor (Sf), as detailed in Section 3.6, expresses the relative substrate specificity of an enzyme (in this case CALB) towards a FA found in a mixture of FAs. The results from Table 7, Table 8 and Table 9 lead to Table 10, in which Sf of CALB and aminoacylases are presented. The Sf of CALB to linoleic acid is the highest in each case, that is, 2.84 in Cu, 2.30 in Rap and 1.36 in Iso-mix. In contrast, in all cases, Sf towards oleic acid was the second highest, and Sf towards palmitic acid was the lowest.

The results obtained with Iso-mix showed that even when the FAs composition is equally separated, the specificity of CALB remains the same as in the cases of Cu and Rap, in which oleic acid was by far the predominant FA. Consequently, it can be said that CALB presents a specificity towards linoleic acid in comparison with the main FAs found in the microbial oil of *C. curvatus* and the vegetable oil of rapeseed.

The same methodology was employed for the N-acylation reaction catalysed by aminoacylases. Two more reconstitutions of FA mixture were prepared as AD carriers, such as a tailor-made *C. curvatus* oil (TMCu) and a tailor-made rapeseed oil (TMRap). These tailor-made FA mixtures allowed us to investigate the potential effect of the hydrolysis of Cu and Rap, due to the hydrolytic specificity of CALB on end products. In Table 9, 2 different major products are presented, depending on the oil or FA mixture used. More specifically, it appears that in HCu, TMCu and Iso-mix α-palmitoyl-lysine was the predominant one, with 59.2%, 53.9% and 65.8% respectively, while in H-Rap and TMRap, it was α-oleoyl-lysine, with 73.1% and 72.2%, respectively. At this point, it is of high importance to consider the FA composition of each oil (% of each AD), in order to understand the substrate specificity of the enzyme, which is expected to verify Iso-mix results, where all acyl donors were in equal concentration and palmitic acid as acyl carrier gave the highest product (α-palmitoyl-lysine).

The specificity factor was calculated considering the real proportions of FAs in the different oils and mixtures. Sf values of aminoacylases are presented in Table 10. No significant differences were noted in the Sf value of the three main FAs between H-Cu and TMCu, as well as between H-Rap and TMRap. Consequently, it can be assumed that the hydrolysis step did not affect the composition of acylation products. Table 10 also demonstrates that the Sf of aminoacylases towards palmitic acid was the highest in all five cases, namely, 2.19, 2.0, 3.06, 2.74 and 1.98. H-Rap, TMRap and Iso-mix oleic acid had the second highest Sf and linoleic acid had the lowest. In contrast, when AD was derived either from H-Cu or TM-Cu, it appears that Sf is almost the same for oleic and linoleic acid. Therefore, it can be concluded that aminoacylases have substrate specificity towards palmitic acid compared to the main FAs found in the microbial oil of *C. curvatus* and the vegetable oil of rapeseed. So far, the substrate specificity of *S. ambofaciens* aminoacylases has been studied only towards isolated acyl acceptors and not in a mixture, namely in the acylation reaction of the twenty proteinogenic amino acids with middle and long-chain FAs [32].

## 3. Materials and Methods

### 3.1. Microorganisms, Enzymes and Chemicals

The yeast strains used in the current study were the *Metschnikowia pulcherima* FMCC Y2, *Rhodotorula* sp. FMCC Y76, *Rhodosporidium toruloides* NRRL Y-6984, *C. curvatus* NRRL Y-1511 and *C. curvatus* ATCC 20509. Strains with the code FMCC Y are indigenous yeast strains that are isolated from foodstuffs [47]. The strains with the code NRRL Y were provided by the NRRL culture collection (Peoria, IL, USA), while the strain with the code ATCC was purchased from the American Type Culture Collection (Manassas, Virginia, USA). The strain *Streptomyces ambofaciens* ATCC 23877, whose aminoacylases were used for biosurfactant synthesis, was also purchased from the American Type Culture Collection (see previously). Maintenance of the strains was conducted as previously indicated [40]. Prior to any inoculation in the liquid growth medium, the strains were regenerated in slants so that the initial slant culture would be that of three days.

*Candida antarctica* lipase B (CALB) enzyme, immobilised on a hydrophobic carrier (acrylic resin), was purchased from Novo Nordisk A/S (Bagsværd, Denmark). All the components of the culture medium of oleaginous yeasts and *S. ambofaciens* were purchased from Sigma-Aldrich (Saint Quentin Fallavier, France), except sucrose and yeast extract which were from Fluka (Munich, Germany). NaCl and CaCO3 were purchased from Carlo Erba (Val-de-Reuil, France). Oleic acid, palmitic acid, linoleic acid, L-Lysine and Tris-HCl were also purchased from Sigma-Aldrich (see previously). Μ2Β_2_ and triethylamine were purchased from Carlo Erba (see previously), and rapeseed oil was purchased from Lesieur (Paris, France). Standards for α-oleoyl lysine and ε-oleoyl-lysine, butanol, acetic acid and methanol were purchased from Sigma-Aldrich (see previously).

### 3.2. Microbial Lipid Production

Liquid cultures were performed in a medium in which the salt composition was as in Papanikolaou et al. [48], while initial glycerol concentration (Glyc0) at concentration ≈ 50 g/L was employed as sole carbon source. Crude glycerol was provided from the Hellenic biodiesel-producing plant “ELIN-VERD SA” (Velestino, Magnesia Prefecture). The purity of the feedstock was ≈85% *w/w*, and the major impurities were composed of salts of potassium and sodium (9%, *w/w*), free-fatty acids (1%, *w/w*) and water (5%, *w/w*). In these media and for all experiments, yeast extract (nitrogen content 11.2% *w/w*) at 1.0 g/L was always added as nitrogen source and as source of vitamins and microelements. In all trials, nitrogen-limited media favouring the accumulation of storage lipids [17,18,19,20] were employed. In these media and besides yeast extract (added, as mentioned, at initial concentration 1.0 g/L), in the screening experiment, peptone (with nitrogen content = 14.2% *w/w*) was also added at initial concentration = 2.0 g/L. Afterwards, and for the case of *C. curvatus* ATCC 20509, which was revealed as the most important lipid-producing microorganism amongst the ones tested, optimisation regarding the nitrogen source was performed. Therefore, peptone (added as mentioned at initial concentration of 2.0 g/L) was replaced by ammonium sulphate (nitrogen content 21.2% *w/w*) at initial concentration = 1.34 g/L, by potassium nitrate (nitrogen content 13.9% *w/w*) at initial concentration = 2.04 g/L, by urea (nitrogen content 46.6% *w/w*) at initial concentration = 0.61 g/L and by yeast extract (as mentioned nitrogen content 11.2% *w/w*) at initial concentration = 2.54 g/L (thus, in the last case, only yeast extract was employed as nitrogen source). Therefore, in all cases, the same absolute quantities of initial glycerol (50 g/L) and nitrogen (0.396 g/L) (initial C/N molar ratio in the media ≈ 57 moles/moles) were employed to identify the most appropriate nitrogen source used in order for SCO to be produced by *C. curvatus* ATCC 20509. In the last set of experiments, with the most appropriate nitrogen source use obtained, trials at higher and lower Glyc0 (e.g., ≈30 and ≈70 g/L) were carried out in order to identify the impact of the Glyc0 upon the process. All cultures were carried out in 250-mL conical non-baffled flasks, filled with the ⅕ of their volume (viz., active volume = 50 ± 1 mL), sterilised (at *T* = 115 °C, 20 min) and inoculated as in Diamantopoulou et al. [40], in an orbital shaker (ZHWY 211C, PR of China) at an agitation rate 180 ± 5 rpm and incubation temperature *T =* 28 ± 1 °C. Initial pH of all trials was =6.0 ± 0.1 and medium pH for all microorganisms irrespective of the culture conditions and the fermentation time ranged between 5.3 and 5.8, with no need for external correction of its value. All experimental data presented (Section 2.1.1) were performed at least in duplicate.

### 3.3. Production of Aminoacylases from Streptomyces ambofaciens

The culture of *Streptomyces ambofaciens* ATCC 23877 was maintained for 7 days at 28 ± 1 °C and 250 ± 5 rpm, pH was set at 7 ± 0.1 before the inoculation, as described by Dettori et al. (2018) [31]. At the end of the fermentation, in order to disrupt the bacterial cells, the entire suspension was subjected 4 times to the Cell disrupter (Constant system Cell-D) at 2.5 kbars. Cellular debris was removed by centrifugation at 8600× *g* at 4 °C for 20 min. The supernatant was then saturated 60% with excess added (NH_4_)_2_SO_4_ and remained for 3 h at room temperature with agitation in order to precipitate proteins, such as aminoacylases. At a later time, the precipitate was recovered by centrifugation at 8600× *g* at 4 °C for 20 min and dissolved in Tris-HCl (25 mM), NaCl (50 mM) buffer, at pH 8. Then, diafiltration was used to concentrate the protein solution and wash out the remaining (NH_4_)_2_SO_4_, 10 kDa ultra filtration membrane was embedded in 3 passages in centrifugation, 5000× *g* at 10 °C for 25 min, after each pass the remaining proteins were dissolved in Tris-HCl (25 mM), NaCl (50 mM) buffer in order to achieve a 10-fold concentration. This *S. ambofaciens* protein crude extract will henceforth be referred to as aminoacylases.

### 3.4. Acylation Reaction

#### 3.4.1. N-Acylation Reaction Catalysed by CALB

Novozym 435^®^ is the lipase B from *Candida antarctica* (CALB) immobilised on a hydrophobic carrier (acrylic resin), with propyl laurate synthesis activity of 7000 PLU/g and protein grade of 1–10% and was used as a biocatalyst for the N-acylation of lysine. The medium consisted of lysine, AD (0.24) and triethylamine (2.4 M) dissolved in 2 mL of Μ2Β2. The latter was stored with dehydrated 4 Å molecular sieves before being added to the medium to reduce the initial water activity below 0.1 and limit the hydrolytic activity of CALB. Triethylamine was used to avoid the protonation of amino groups. The N-acylation reaction was initiated by the addition of CALB (10 g/L), which, due to the mass of the enzyme immobilisation material, corresponds to about 1 g/L of enzyme. The reaction medium was stirred from 400 ± 5 to 500 ± 5 rpm and maintained at 55 ± 1 °C [15,28], 50 μL samples were withdrawn over time for analyses, diluted with methanol/water (80/20, *v/v*) and then stored at room temperature until analytical methods were performed. Each reaction was repeated at least twice, the results are presented as average with standard deviation.

#### 3.4.2. N-Acylation Reaction Catalysed by Aminoacylases

N-acylation reaction by aminoacylases derived from *S. ambofaciens* culture, as described in Section 3.3, took place in an aqueous medium. The latter consisted of lysine (0.1 M) and AD (0.1 M) dissolved in 2 mL of Tris-HCl (25 mM), NaCl (50 mM) buffer (24). The N-acylation reaction was initiated by the addition of aminoacylases (1 g/L), while the reaction medium was stirred from 500 ± 5 to 700 ± 5 rpm, pH was set at 8 and temperature was maintained at 45 ± 1 °C. In the case of rapeseed and *C. curvatus* oil, a hydrolysis step was required prior to the acylation reaction. Its medium consisted of oil (0.24 M) and CALB (10 g/L) dissolved in 5 ml of Tris-HCl (25 mM), NaCl (50 mM) buffer. To favour the hydrolytic activity of CALB, biocatalysis took place at 37 ± 1 °C, pH 8 and 550 ± 5 rpm for 24 h. The medium was then centrifuged at 9000× *g* at 15 °C for 10 min and the supernatant was filtrated with a 0.2 μm membrane to remove CALB. Next, aminoacylases and lysine were added to the medium. Over time, 50 μL samples were withdrawn for analyses and were diluted with methanol/water (80/20, *v/v*) and then stored at room temperature until analytical methods were performed. Each reaction was repeated at least twice, the results are presented as average with standard deviation.

### 3.5. Analytical Methods

#### 3.5.1. Microbial Lipid Production

The whole content of the flasks (≈46–48 mL) was collected in order to perform various analyses as regards the production of biomass, the assimilation of substrate and the biosynthesis of the extra- and intra-cellular metabolites. Due to slight water evaporation, the volume of liquid medium in the flasks after harvesting was 47 ± 1 mL. This volume was always corrected to 50 mL and the whole content of the flask was used in the analyses performed. Yeast cells were harvested by centrifugation, and yeast total biomass quantity was determined through its DCW [35,40]. Extra-cellular compounds (viz., glycerol) were quantitatively determined by High-Pressure Liquid Chromatography (HPLC) analysis as previously reported [40]. Cellular lipids were extracted with the modified “Folch” method, as described in Sarantou et al. (2021), where dried biomass was put in a McCartney vial and was covered with chloroform/methanol (C/M) 2:1 (*v/v*) blend [35]. The hole was closed with aluminium screw cap and was left for at least 6 days in the darkness. Occasionally, the content of the vial was gently mixed with the aid of a glass stick. Cell debris was removed through filtration (Whatman^®^ filter n° 3) and the solvent mixture was collected in a pre-weighted evaporator flask and was completely evaporated in a rotary evaporator (R-144, Büchi Labortechnik, St. Gallen, Switzerland), with total lipid being determined gravimetrically. After the recovery, total lipids were trans-methylated and their total FA composition was evaluated by Gas Chromatography (GC) analysis [35]. Rapeseed oil was also analysed by GC analysis for its total fatty acid composition. Intra-cellular total polysaccharides were quantitatively determined according to Filippousi et al. (2022) [36]. Finally, dissolved oxygen concentration (DOC) in the flask cultures was measured offline with a selective electrode (HI 9146, Hanna Instruments, Woonsocket, RI, USA) as previously indicated in Filippousi et al. (2022) [36]. In all experiments, for all strains and irrespective of the culture conditions, trials were performed under fully aerobic conditions, with DOC being always ≥30% *v/v*.

#### 3.5.2. Protein Concentration Measurement

Final protein concentration of aminoacylases was assayed according to the Bradford spectrophotometric method [49], using bovine serum as standard. Both aminoacylases samples and standards were added to pre-formulated Coomassie blue G-250 assay reagent and the resultant blue colour is measured at 595 nm following a short room temperature incubation.

#### 3.5.3. Quantitative Analysis 

The quantification of products (N-acylated-lysine) was conducted using HPLC (LC10 AD-VP, Shimadzu, Noisiel, France) equipped with a UV detector at 214 nm and a light-scattering low-temperature evaporative detector (Shimadzu, Noisiel, France). A C18 amide 125 × 2.1 mm (Altima^®^, Altech, Vire-Normandie, France) maintained at 25 °C was used as the suitable separation column. The mobile phase (0.2 mL/min flow rate) consisted of solvent A: methanol/water/TFA (60/40/0.1, *v/v/v*) and solvent B: methanol/TFA (100/0.07, *v/v*). Starting from solvent A 95% and solvent B 5%, a linear elution gradient was applied to reach 95% methanol/TFA after 6 min. This methanol concentration was maintained for 13 min and then decreased in 1 min to reach the initial methanol/water ratio until the end of the run (35 min). Calibrations were performed using standard curve FAs and purified commercial α- and ε-oleoyl-lysine [31].

#### 3.5.4. Qualitative and Semi-Quantitative Analysis by Mass Spectrometry

The synthesis of lysine derivatives was monitored with a Thermo Scientific (Waltham, MA, USA) UHPLC-HRMS/MS system composed of a Vanquish™ liquid chromatography unit coupled to a photodiode array detector (PDA) and an Orbitrap ID-X™ Tribrid™ high-resolution mass spectrometer operating in electrospray mode (ESI). Chromatographic separation was performed on an Alltima C18 column (100 × 2.1 mm, 3 μm granulometry, Hichrom, Reading, England) equipped with a C18 pre-column (7.5 × 2.1 mm, 3 μm granulometry, Hichrom, Reading, England) at 30 °C. Mobile phases consisted of methanol/water/TFA (80/20/0.1, *v/v/v*) for phase A and methanol/TFA (100/0.1, *v/v*) for phase B. Elution was performed using a linear gradient from 0% to 98% of B for 10 min and then an isocratic step at 98% of B for 5 min, at a flow rate of 0.2 mL/min. Mass analysis was carried out in ESI positive ion mode (ESI^+^) and mass spectrometry conditions were as follows: spray voltage was set at 3.5 kV; source gases were set (in arbitrary units·min^−1^) for sheath gas, auxiliary gas and sweep gas at 35, 7 and 10, respectively; vaporiser and ion transfer tube temperatures were both set at 300 °C. Survey scans of precursors were performed from 150 to 2000 *m/z* at 7.5 K resolution (FWHM at 200 *m/z*) with MS parameters as follows: RF-lens, 35%; maximum injection time, 50 ms; data type, profile; normalised AGC target: 25%. A top speed data-dependent MS^2^ (0.6 sec for the whole cycle time) was carried out using a wide quadrupole isolation (1.5 Th), an HCD fragmentation with stepped collision energy (20%, 35% and 50%) and an Orbitrap measure at 7.5 K resolution. Precursors with an intensity greater than 2.10^4^ were automatically sampled for MS^2^. Dynamic exclusion was used and the time of exclusion was set at 2.5 s, with a 10 ppm tolerance around the selected precursor (isotopes excluded). Other MS^2^ parameters were as follows: data type, profile; normalised AGC target: 20%; AGC target, 10000. MS data acquisition was carried out by using the Xcalibur v. 3.0 software (Thermo Scientific).

### 3.6. Specificity Factor (Sf) Formula

Specificity factor (Sf) can be determined, based on the masses area of products in HPLC-MS and the FA composition of mixture used. Sf expresses the ratio between the % of a product in a sum of products and the % of the AD, from which the product is derived, in the oil or the mixture (Equation (1)).

More precisely, it is used to describe the relative substrate specificity of an enzyme (e.g., CALB, aminoacylases) to a FA found (e.g., oleic acid, linoleic acid, palmitic acid) in a mixture of FAs (e.g., *C. curvatus* oil, Rapeseed oil).
(1)$$\mathrm{Sf}=\frac{\%\;\mathrm{of}\;\mathrm{Product}\;\mathrm{in}\;\mathrm{the}\;\mathrm{sum}\;\mathrm{of}\;\mathrm{products}}{\%\;\mathrm{of}\;\mathrm{AD}\;\mathrm{in}\;\mathrm{the}\;\mathrm{oil}\;\mathrm{or}\;\mathrm{in}\;\mathrm{the}\;\mathrm{mixutre}}$$

Sf: Specificity factor, AD: Acyl donor.

### 3.7. Tailor-Made FA mixtures

The aim of tailor-made FA mixture preparation was to study the different substrate specificities of the enzymes. More specifically, both the FAs form (i.e., triacylglycerol, hydrolysed triacylglycerols, free FAs), and the level of their concentration in the mixture was studied for the effect that they might have on the substrate specificity of an enzyme.

Three FA mixtures were prepared, namely, tailor-made *C. curvatus* major FAs (TMCu), tailor-made rapeseed oil major FAs (TMRap) and an equimolar mixture containing the major FAs of both oils (Iso-mix). Therefore, TMCu and TMRap contained the three major FAs in the same ratio found in *C. curvatus* and rapeseed oil, respectively. The reason for their preparation is to study the different effects between the free FAs of a tailor-made mixture and a hydrolysed oil on the substrate specificity of aminoacylases. Iso-mix is an equimolar mixture of the three major FAs of *C. curvatus* and rapeseed oil, namely palmitic acid, oleic acid and linoleic acid. In this case, the aim was to study the effect of FA molar ratio on the substrate specificity of both CALB and aminoacylases. First, the concentrations of each FA in each mixture were calculated according to the results of oils analysed by GC and then each part was weighed and mixed.

### 3.8. Abbreviations

Table 11 provides the meaning of the abbreviations used to describe different FA mixtures and oils.

## 4. Conclusions

As far as we know, this is the first time reported in the contemporary literature that a microbial oil is used as AD for biosurfactant synthesis. This biocatalytic reaction was successfully performed by two enzymes, aminoacylases and CALB (as reference enzyme). The bioprocess presented, involving the fermentation of *C. curvatus* for the production of SCO, the fermentation of *S. ambofaciens* for the production of aminoacylases and the enzymatic synthesis of amino acid-based biosurfactants from SCO, took place in a fully aqueous environment.

New findings on aminoacylase ability, substrate specificity and enzymatic performance are presented. More precisely, it is for the first time stated that aminoacylases can perform N-acylation only when free FAs are used as ADs, hence a microbial oil should first be hydrolysed. Based on the new parameter, viz., Sf, aminoacylases have a substrate specificity towards palmitic acid, compared to oleic and linoleic acid. Aminoacylases catalyse the N-acylation reaction of lysine and produce amino acid biosurfactants from SCO in aqueous mediums and present higher productivity and product titre than with vegetable oil, which is a food-competitive “feedstock”. Finally, CALB proved to have substrate specificity towards linoleic acid compared to oleic acid and palmitic acid. Understanding such characteristics of an enzyme could significantly contribute to the development of new highly customised bioprocesses, with maximum productivity and selectivity. Further work is underway to develop continuous coupled microbial-enzymatic bioprocesses and solvent-free extraction methods of microbial oils.

## Figures and Tables

**Figure 1 ijms-24-00714-f001:**
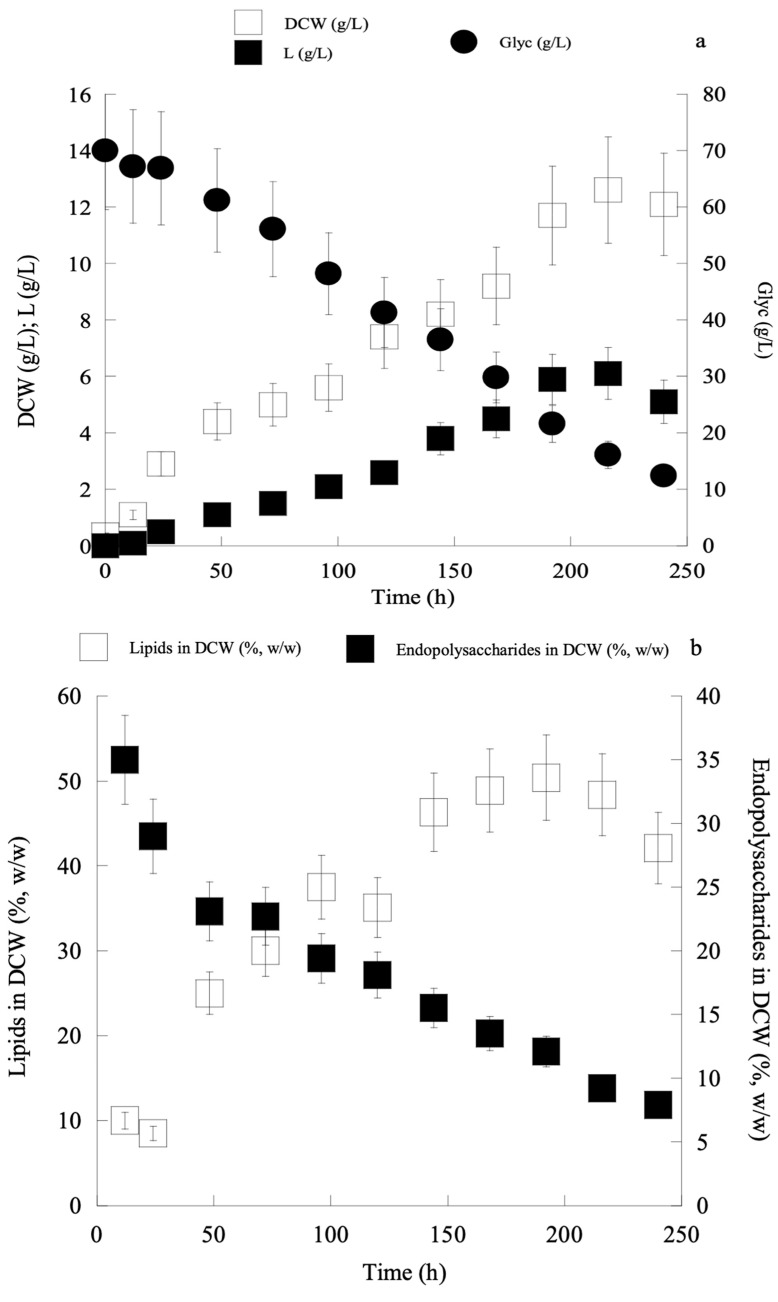
Kinetics of dry cell weight (DCW, g/L) (**a**), lipid (L, g/L) (**a**), glycerol (Glyc, g/L) (**a**), quantity of lipid per DCW (%, *w/w*) (**b**), quantity of intra-cellular polysaccharides per DCW (%, *w/w*) (**b**) by *Cryptococcus curvatus* ATCC 20509, when cultivated on crude glycerol in shake-flask experiments, Glyc0 ≈ 70 g/L; C/N ≈ 80 moles/moles.

**Figure 2 ijms-24-00714-f002:**
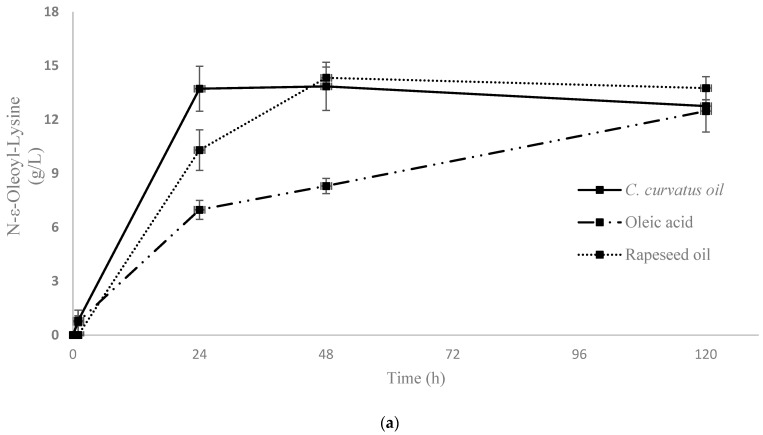
Kinetics of enzymatic synthesis of ε and α oleoyl-lysine by CALB (**a**) and aminoacylases (**b**), respectively, in different fatty acid mixtures and oils. Two replicates are presented in both diagrams.

**Figure 3 ijms-24-00714-f003:**
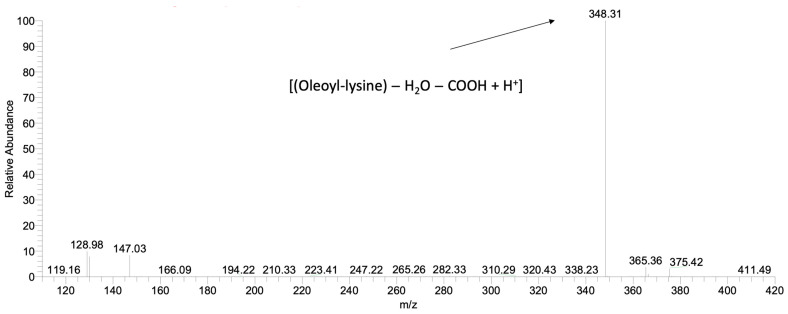
HPLC-MS2 spectrum after fragmentation of the ε-oleoyl-lysine parent ion (*m/z* = 410) synthesised by CALB.

**Figure 4 ijms-24-00714-f004:**
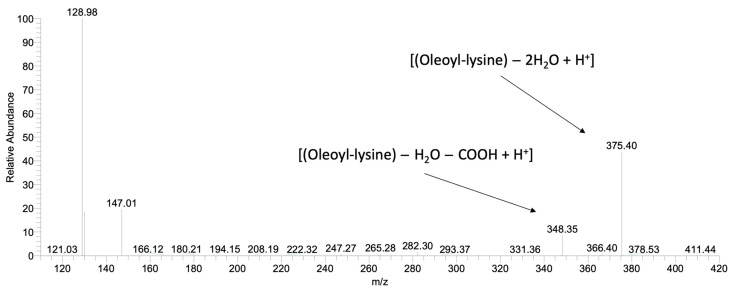
HPLC-MS2 spectrum after fragmentation of the α-oleoyl-lysine, parent ion (*m/z* = 410), synthesised by aminoacylases.

**Table 1 ijms-24-00714-t001:** Quantitative data of *Metschnikowia pulcherrima*, *Rhodotorula* sp., *Rhodosporidium toruloides* and *Cryptococcus curvatus* strains (a total of 5 strains) deriving from kinetics on crude glycerol, in nitrogen-limited shake-flask cultures, with initial glycerol (Glyc_0_) concentration ≈ 50 g/L. Four different points in the fermentations are represented: (point 1) when the maximum quantity of DCW (X, g/L) was observed; (point 2) when the maximum quantity of lipid per DCW (Y_L/X_, g/g) was observed; (point 3) when the maximum quantity of absolute lipid value (L, g/L) was observed; (point 4) when the maximum quantity of intra-cellular polysaccharides per DCW (Y_IPS/X_, g/g) was observed. Fermentation time (h) and glycerol consumed (Glyc_cons_, g/L) are also depicted for all the above-mentioned fermentation points. Culture condition: growth on 250-mL conical flasks at 180 ± 5 rpm, initial pH = 6.0 ± 0.1, culture pH ranging between 5.2 and 5.8, incubation temperature *T* = 28 °C. Each experimental point is the mean value of two measurements (SE < 15%).

Strains		Time (h)	Glyc_cons_ (g/L)	X (g/L)	L (g/L)	Y_L/X_ (g/g)	Y_IPS/X_ (g/g)
*Metschnikowia*	2	146	35.5	11.0	0.61	0.06	0.18
*pulcherrima* FMCC Y2	1, 3, 4	288	47.9	16.5	0.85	0.05	0.43
*Cryptococcus*	4	162	36.2	15.5	0.26	0.02	0.25
*curvatus*	1	186	40.7	17.6	0.36	0.02	0.19
NRRL Y-1511	2, 3	211	45.5	16.6	0.95	0.06	0.14
*Cryptococcus*	4	47	14.8	7.9	0.91	0.11	0.41
*curvatus*	1	119	30.5	15.9	3.78	0.24	0.32
ATCC 20509	2, 3	201	47.9	15.1	4.98	0.33	0.19
*Rhodosporidium*	4	48	11.9	7.2	0.74	0.10	0.31
*toruloides*	1	191	43.2	16.4	3.18	0.19	0.17
NRRL Y-6984	2, 3	215	43.2	16.0	3.23	0.20	0.16
*Rhodotorula* sp.	2, 3	210	31.9	15.8	2.60	0.16	0.36
FMCC Y76	1, 4	255	36.9	17.2	1.91	0.11	0.39

**Table 2 ijms-24-00714-t002:** Quantitative data of *Cryptococcus curvatus* strain ATCC 20509 deriving from kinetics on crude glycerol, in nitrogen-limited shake-flask cultures, in which the initial glycerol (Glyc0) was ≈50 g/L, and the same initial quantity of nitrogen was imposed, with different nitrogen sources employed (in all trials, yeast extract at an initial concentration of 1.0 g/L has already been employed). Four different points in the fermentations are represented: (point 1) when the maximum quantity of DCW (X, g/L) was observed; (point 2) when the maximum quantity of lipid per DCW (Y_L/X_, g/g) was observed; (point 3) when the maximum quantity of absolute lipid value (L, g/L) was observed; (point 4) when the maximum quantity of intra-cellular polysaccharides per DCW (Y_IPS/X_, g/g) was observed. Fermentation time (h) and quantity of glycerol consumed (Glyc_cons_, g/L) are also depicted for all the above-mentioned fermentation points. Culture condition: growth on 250-mL conical flasks at 180 ± 5 rpm, initial pH = 6.0 ± 0.1, culture pH ranging between 5.2 and 5.8, incubation temperature *T* = 28 °C. Each experimental point is the mean value of two measurements (SE < 15%).

Nitrogen Source		Time (h)	Glyc_cons_ (g/L)	X (g/L)	L (g/L)	Y_L/X_ (g/g)	Y_IPS/X_ (g/g)
Ammonium	4	71	14.8	13.6	1.69	0.12	0.27
sulfate	2, 3	144	30.5	18.5	3.35	0.18	0.21
(1.34 g/L)	1	192	44.8	20.0	2.90	0.15	0.11
Peptone	4	47	14.8	7.9	0.91	0.11	0.41
(2.00 g/L)	1	119	30.5	15.9	3.78	0.24	0.32
	2, 3	201	47.9	15.1	4.98	0.33	0.19
Potassium	4	47	8.3	6.0	0.79	0.13	0.40
nitrate	2, 3	192	29.1	11.6	4.52	0.39	0.09
(2.04 g/L)	1	206	29.7	12.0	3.92	0.33	0.08
Urea	3	46	8.8	6.8	1.07	0.16	0.26
(0.61 g/L)	4	96	18.3	11.4	1.01	0.09	0.44
	1, 2	170	24.7	14.8	1.90	0.12	0.36
Yeast	4	72	12.6	14.0	2.17	0.16	0.36
extract	2, 3	170	29.4	21.2	3.16	0.25	0.20
(2.54 g/L)	1	216	31.1	21.4	3.08	0.24	0.18

**Table 3 ijms-24-00714-t003:** Quantitative data of *Cryptococcus curvatus* strain ATCC 20509 deriving from kinetics on crude glycerol, in nitrogen-limited shake-flask cultures, in which various initial glycerol (Glyc0) concentrations (≈30, ≈50 and ≈70 g/L) and the same initial quantity of nitrogen (yeast extract at 1.0 g/L and peptone at 2.0 g/L) were imposed. Media with 3 different initial molar ratios C/N (≈34, ≈57 and ≈80 moles/moles) were created. Four different points in the fermentations are represented: (a) when the maximum quantity of DCW (X, g/L) was observed; (b) when the maximum quantity of lipid per DCW (Y_L/X_, g/g) was observed; (c) when the maximum quantity of absolute lipid value (L, g/L) was observed; (d) when the maximum quantity of intra-cellular polysaccharides per DCW (Y_IPS/X_, g/g) was observed. Fermentation time (h) and quantity of glycerol consumed (Glyc_cons_, g/L) are also depicted for all the above-mentioned fermentation points. Culture condition: growth on 250-mL conical flasks at 180 ± 5 rpm, initial pH = 6.0 ± 0.1, culture pH ranging between 5.2 and 5.8, incubation temperature *T* = 28 °C. Each experimental point is the mean value of two measurements (SE < 15%).

Glyc0 (g/L)	Initial C/N (moles/moles)		Time (h)	Gly_cons_ (g/L)	X (g/L)	L (g/L)	Y_L/X_ (g/g)	Y_IPS/X_ (g/g)
		4	45	13.9	5.1	0.88	0.17	0.39
≈30 g/L	≈34	2, 3	120	29.5	14.0	2.71	0.19	0.31
		1	155	34.0	16.1	1.95	0.12	0.27
		4	47	14.8	7.9	0.91	0.11	0.41
≈50 g/L	≈57	1	119	30.5	15.9	3.78	0.24	0.32
		2, 3	201	47.9	15.1	4.98	0.33	0.19
		4	12	2.9	1.1	0.11	0.10	0.35
≈70 g/L	≈80	2	192	48.4	11.7	5.90	0.50	0.12
		1, 3	216	57.4	12.6	6.10	0.48	0.09

**Table 4 ijms-24-00714-t004:** Fatty acid composition of the cellular lipids produced by yeast strains cultivated on crude glycerol in shake-flask experiments (Glyc0 ≈ 50 g/L). Time of fermentation for the determination of the fatty acid composition was between 170 and 220 h after inoculation. Culture conditions as in Table 1. Tr.: traces (<0.5%).

	Fatty Acid Composition of Yeast Lipids (%, *w/w*)					
Yeast Strain	C16:0	C16:1	C18:0	C18:1	C18:2	C18:3
*Metschnikowia pulcherrima* FMCC Y2	17.7	7.7	Tr.	59.4	14.5	Tr.
*Cryptococcus curvatus* NRRL Y-1511	33.8	5.6	0.7	45.2	11.7	3.0
*Cryptococcus curvatus* ATCC 20509	26.0	1.2	3.9	57.2	11.2	Tr.
*Rhodosporidium toruloides* NRRL Y-6984	33.9	0.5	1.2	56.7	7.8	Τr.
*Rhodotorula* sp. FMCC Y76	21.0	0.5	7.5	51.0	12.1	1.0

**Table 5 ijms-24-00714-t005:** Fatty acid composition of the cellular lipids produced by *Cryptococcus curvatus* ATCC 20509, when cultivated on crude glycerol in shake-flask experiments (Glyc0 ≈ 50 g/L), when various nitrogen sources were used. Time of fermentation for the determination of the fatty acid composition was between 170 and 220 h after inoculation. Culture conditions as in Table 2. Tr.: traces (<0.5%).

	Fatty Acid Composition of Yeast Lipids (%, *w/w*)					
Nitrogen Source	C16:0	C16:1	C18:0	C18:1	C18:2	C18:3
Ammonium sulfate	37.2	Tr.	Tr.	49.4	13.0	Tr.
Potassium nitrate	35.6	6.8	Tr.	54.5	3.2	Tr.
Peptone	26.0	1.2	3.9	57.2	11.2	Tr.
Urea	34.3	3.6	Tr.	47.4	8.4	Τr.
Yeast extract	39.2	4.2	Tr.	54.9	8.8	0.5

**Table 6 ijms-24-00714-t006:** Fatty acid composition of the cellular lipids produced by *Cryptococcus curvatus* ATCC 20509, when cultivated on crude glycerol in shake-flask experiments and when various initial glycerol concentrations were used. Time of fermentation for the determination of the fatty acid composition was between 170 and 220 h after inoculation. Culture conditions as in Table 2. Tr.: traces (<0.5%).

	Fatty Acid Composition of Yeast Lipids (%, *w/w*)					
Glyc0 (g/L)	C16:0	C16:1	C18:0	C18:1	C18:2	C18:3
≈30 g/L	30.2	3.5	4.2	48.7	12.7	Tr.
≈50 g/L	26.0	1.2	3.9	57.2	11.2	Tr.
≈70 g/L	27.0	1.5	3.0	45.0	9.5	2.0

**Table 7 ijms-24-00714-t007:** The major fatty acid composition of oils and mixture used for the N-acylation of lysine.

	Major Fatty Acid Composition (%, *w/w*)		
	C16:0	C18:1	C18:2
*C. curvatus* oil	27.0	45.0	9.5
Rapeseed oil	4.8	60.0	20.0
Iso-mix	33.3	33.3	33.3

**Table 8 ijms-24-00714-t008:** Enzymatic acylation by CALB, Product (Mass Area) referred to the N-ε-acylation of lysine by an AD, Product (%), is the percentage of each product in the sum of 3 main products that occurred in each oil or FA mixture. HPLC-MS analysis provided Mass Area M + H^+^.

	ε-Palmitoyl-Lysine		ε-Oleoyl-Lysine		ε-Linoleoyl-Lysine	
	Product (Mass Area)	Product (%)	Product (Mass Area)	Product (%)	Product (Mass Area)	Product (%)
*C. curvatus* oil	2.8	6.8	27.5	66.3	11.2	27
Rapeseed oil	1	2.8	18.8	51.2	16.9	46
Iso-mix	6	20.3	10.1	34.4	13.3	45.3

**Table 9 ijms-24-00714-t009:** Enzymatic acylation by aminoacylases, Product (Mass Area) referred to the N-α-acylation of lysine by an AD, Product (%), is the percentage of each product in the sum of 3 main products that occurred in each oil or FA mixture. HPLC-MS analysis provided Mass Area M + H^+^.

	α-Palmitoyl-Lysine		α-Oleoyl-Lysine		α-Linoleoyl-Lysine	
	Product (Mass Area)	Product (%)	Product (Mass Area)	Product (%)	Product (Mass Area)	Product (%)
H-*C. curvatus* oil	26.3	59.2	14.9	33.6	3.17	7.1
TM-*C. curvatus*	27.1	53.9	19.2	38.2	4	7.9
H-Rapeseed oil	3.4	14.7	16.8	73.1	2.8	12.2
TM-Rapeseed	3.9	13.1	21.6	72.2	4.4	14.6
Iso-mix	28.9	65.8	10.3	23.5	4.7	10.7

**Table 10 ijms-24-00714-t010:** Specificity factor of CALB and aminoacylases towards the 3 major fatty acids of oils and tailor-made mixtures.

	C16:0 (Sf)	C18:1 (Sf)	C18:2 (Sf)
CALB			
*C. curvatus* oil	0.25	1.47	2.84
Rapeseed oil	0.57	0.85	2.30
Iso-mix	0.61	1.03	1.36
Aminoacylases			
H-*C. curvatus* oil	2.19	0.75	0.75
TM-*C. curvatus*	2.00	0.85	0.84
H-Rapeseed oil	3.06	1.22	0.61
TM-Rapeseed	2.74	1.20	0.73
Iso-mix	1.98	0.71	0.32

**Table 11 ijms-24-00714-t011:** Abbreviation and meaning for fatty acids mixtures and oils.

Abbreviation	Meaning
Cu	Oil derived from *Cryptococcus curvatus* ATCC 20509 culture
H-Cu or H-*C. curvatus* oil	Hydrolysed *Cryptococcus curvatus* oil
TMCu or TM-*C. curvatus*	Tailor-made solution of the major fatty acids of *Cryptococcus curvatus* oil
Rap	Rapeseed oil
H-Rap or H-Rapeseed oil	Hydrolysed rapeseed oil
TMRap or TM-Rapeseed	Tailor-made mixture of the major fatty acids of rapeseed oil
Iso-mix	Tailor-made solution containing 1/3 palmitic acid, 1/3 oleic acid and 1/3 palmitic acid

## Data Availability

Not applicable.

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
