# Peer review of "Yeast Lipid Produced through Glycerol Conversions and Its Use for Enzymatic Synthesis of Amino Acid-Based Biosurfactants"

_ijms, 2022, doi:10.3390/ijms24010714_

Round 1

Reviewer 1 Report

The work described in this article is topical of great industrial relevance. The paper describes a big leap forward in using single cell oil to make biosurfactant in good yield. I have certain comments that should be taken into consideration for further improvement during revision of the manuscript. The conclusions are supported by the data. It is very well written. However, there are lots of small English/technical mistakes that needs to be corrected. I have listed few of them below. There are only 3 major comments (14-16) that must be taken into account and discussed in the revised manuscript.

L1.Line 42: should also include microalgae

22. Line 74: needs rewriting. resulted in a steep increase in the cost of several types of agro-industrial / agricul-

33.       Line 79: replacements for the mentioned plant-based fatty materials…

44.       Line 84: and provide elevated reaction rates, high specificity and regioselec-

55.       Line 90: to perform lysine acylation in of ε-amino group in organic solvent

66.       Line 104: as ε-lauroyl-lysine yield was 10-fold higher than α-lauroyl-lysine.

77.       Line 123: observed when acylation occurred in at α-position.

88.       Section 2.4.2: Why the temperatures of Lipase and amino acylase reactions are different? That will make it difficult to compare them. Also, the enzyme concentrations for both enzymes are very different. Is the lipase concentration used (10 g/L) corresponds to immobilized enzyme? Give actual lipase concentration per 10 g immobilized resin.

99.       Section 2.5.5: Was any internal standard used for normalizing MS data?

110.   Line 336: Rewrite. it was studied the effect that

111.   Line 343: reason for their addition existence is to study the

112.   Do not use the symbol “Gly” for glycerol because Gly is universally known as a symbol for glycine amino acid. Use Glyc of Gro.

113.   How Figure 3 comes before Figure 2? Correct.

114.   IMPORTANT MAJOR COMMENTS: Fig 2a & b and the related section about productivity needs further explanation/discussion. This is because productivity results are the key to this paper. Kindly rewrite discussion about productivity analysis in view of the following recent paper (Int. J. Mol. Sci. 202223(13), 6908; https://doi.org/10.3390/ijms23136908). Fig 2a shows that for all three oils the maximum productivity was reached at different times. This could also be due to lipase being unfolded earlier in that oil or due to the product inhibition or that all the oil being consumed earlier.

115.   In Fig 2b, the productivity increased slowly up to 120 h for two top oils. The aim for industrial application is to attain maximal productivity in less time to be cost effective. Here increasing temperature or enzyme load could further decrease time and increase productivity (g/L/h). Also comment in view of the published paper suggested above what could be the reason for achieving the plateau after 120 h? Is it total consumption of substrate, enzyme loss due to unfolding or product inhibition?   

116.   Lines 575-580, Section 3.2.2: How the free fatty acids obtained? Comment what would happen if Lipase B were used to hydrolyse oil in combination with aminoacylase crude extract present together? Would that be an advantage?

Author Response

Please see the attachment "Reviewer 1"

Reviewer 2 Report

Dear Editor,

In this manuscript, Karayannis and co-authors screened five oleouginous yeasts to select the best lipid producer. Then single cell oil production was optimized and lipids were extracted and used as substrate in acylation reactions to obtain amino acid-based surfactants.

Overall the manuscript is well-written and the conclusions are supported by the experimental results.

Analytical comments

Abstract: the abstract is too detailed and full of experimental data, I suggest to reorganize it as follows: background, scope of the work, main results and conclusion.

Material and methods

- It is unclear how many replicates the authors made. Please provide this information.

Line 172. It is not clear whether the authors used free or immobilized CALB to perform the reactions. Please specify.

Results

- Tables 2 and 3 are cryptic and difficult to read, especially the lanes corresponding to the letters "a-b".

- Figure 1 is too big, the letters (a) and (b) should be moved to the upper left corner. In addition error bars are missing.

- Please check the numbers of the figures. Currently Fig. 3 precedes Fig. 2.

- The mass spectra in Fig. 3 and 4 should be moved into the supplementary materials. 

- Please indicate the number of replicates (n) in figure 2.

- This work contains 11 tables, making the manuscript difficult to read. I wonder if the results of some tables can be shown in column plots.

Author Response

Please see the attachment "Reviewer 2".
